# A general theoretical framework to design base editors with reduced bystander effects

Qian Wang [1,2,7 ✉], Jie Yang [3,7], Zhicheng Zhong[1], Jeffrey A. Vanegas [3], Xue Gao [3,4,5 ✉] & Anatoly B. Kolomeisky [2,3,5,6 ✉]

Base editors (BEs) hold great potential for medical applications of gene therapy. However, high precision base editing requires BEs that can discriminate between the target base and multiple bystander bases within a narrow active window (4 – 10 nucleotides). Here, to assist in the design of these optimized editors, we propose a discrete-state stochastic approach to build an analytical model that explicitly evaluates the probabilities of editing the target base and bystanders. Combined with all-atom molecular dynamic simulations, our model reproduces the experimental data of A3A-BE3 and its variants for targeting the "T<u>C</u>" motif and bystander editing. Analyzing this approach, we propose several general principles that can guide the design of BEs with a reduced bystander effect. These principles are then applied to design a series of point mutations at T218 position of A3G-BEs to further reduce its bystander editing. We verify experimentally that the new mutations provide different levels of stringency on reducing the bystander editing at different genomic loci, which is consistent with our theoretical model. Thus, our study provides a computational-aided platform to assist in the scientifically-based design of BEs with reduced bystander effects.

---

[1] Hefei National Laboratory for Physical Sciences at the Microscale and Department of Physics, University of Science and Technology of China, Hefei 230026 Anhui, China. [2] Center for Theoretical Biological Physics, Rice University, Houston, TX 77005, USA. [3] Department of Chemical and Biomolecular Engineering, Rice University, Houston, TX 77005, USA. [4] Department of Bioengineering, Rice University, Houston, TX 77005, USA. [5] Department of Chemistry, Rice University, Houston, TX 77005, USA. [6] Department of Physics and Astronomy, Rice University, Houston, TX 77005, USA. [7] These authors contributed equally: Qian Wang, Jie Yang. ✉email: wqq@ustc.edu.cn; xue.gao@rice.edu; tolya@rice.edu

The development of genome editing tools associated with the clustered regularly interspaced short palindromic repeat (CRISPR) systems has revolutionized biomedical studies. Holding great potential for the treatment of genetic diseases, diverse precise genome editing tools based on CRISPR-Cas9 have been developed, such as homology-directed repair (HDR) based systems, as well as cytosine and adenine base editors (BE)[1–3]. While the HDR method requires double-stranded DNA breaks (DSBs) and causes unpredictable editing outcomes, BEs use nickase Cas9 (nCas9), enabling more precise modifications without generating DSBs[4–6]. For example, cytosine BEs (CBEs) are constructed by fusion of a cytidine deaminase domain with nCas9. This fusion protein forms a complex with the guide RNA and performs site-specific deamination to convert cytosine (C) to uracil (U) in the deaminase activity window. The base U is subsequently replaced with thymine (T) by the endogenous cellular repairing machinery, resulting in an overall C-to-T substitution at the defined genomic site. Since point mutations are responsible for more than half of human disease-associated genetic variants[2], BEs are superior than HRD based systems due to their higher editing efficiency in the correction of pathogenic single nucleotide polymorphisms[7], avoidance of unwanted DSBs, and prevention of insertions and deletions[4,5].

While engineering of several BE variants has improved product purity and overall editing efficiency[8,9], one of the major challenges in base editing is the discrimination of multiple identical bases located within the deaminase activity window[2] of 4–10 nucleotides. As a result, the target base and other bystander bases will all be modified, negatively impacting the precision of genome editing outcomes. To address this issue, the introduction of beneficial mutations to deaminase has further advanced BEs[10–12]. For example, compared to the wild-type APOBEC3A (A3A)-BE3, an engineered A3A CBE with the mutation N57G maintained high editing activity at the target C in the TCR motif with greatly reduced activity against bystanders[11]. Also, followed by several rounds of screening and validation of rational mutagenesis, we previously engineered an APOBEC3G (A3G)-CBE that preferentially edits the second C in the "CC" motif with 6000-fold improvement in perfectly modified alleles compared to the original BE4max[12]. Despite these successes, a general theoretical framework to guide the design of mutations that can lead to high editing activity at the target base and low activity at bystanders (defined as BE high editing selectivity) is still missing. Mutation selections in these previous studies were mostly suggested by structural considerations: starting from the identification of key residues in the deaminase near the DNA binding motif and then mutating those residues to form a candidate library for experimental validation. The design process could be greatly accelerated with a comprehensive theoretical model that could quantitatively explain and predict the effect of specific mutations on editing activity at the target base and bystanders. In addition, such a theoretical model would also improve our fundamental understanding of the biochemical and biophysical processes that take place during base editing.

Molecular dynamic (MD) simulations have been used to study the activity of BE complexes and the role of beneficial mutations to enhance overall editing activity (both at target and bystanders)[13,14]. Herein we present a comprehensive multi-scale theoretical approach to describe the molecular processes taking place during BE editing, explaining at the microscopic level the role of beneficial mutations in discriminating the target base over bystanders. To fulfill this goal, we built a general theoretical framework combining a discrete-state stochastic (chemical-kinetic) model and MD simulations, explicitly calculating the base editing probability at both the target base and bystanders. In our model, we include an important parameter, $\triangle E_m$, the binding affinity between deaminases and ssDNA. This parameter was modulated by introducing various mutations into BE and its values were measured through MD simulations. This framework helps establish a relationship between mutations and BE editing selectivity. We then proposed a theoretical principle arguing that the BE selectivity is non-monotonically dependent on $\triangle E_m$. It is argued that the highest BE selectivity can be obtained by varying the binding affinity. In addition, other relevant kinetic parameters are included in the model, such as the binding rate between Cas9 and ssDNA and the deamination rate of BE, allowing us to discuss how $\triangle E_m$ correlates with those parameters to affect BEs editing selectivity. Our model successfully explains how the mutations influence the editing selectivity of A3A-BE3. Finally, we designed mutations to further improve the selectivity of the A3G-BE system and we verified the improved editing selectivity experimentally. Thus, the framework we propose opens multiple opportunities for future efficient engineering of BE using theory-driven methods.

## Results

**Kinetic model of base editing.** We developed a discrete-state stochastic model to describe the dynamics of target and bystander editing. This is a minimal chemical-kinetic approach that considers the most relevant chemical states and features of base editing. For convenience, unless noted otherwise, we will use A3A-BE3 to edit the EGFP site 1 as an example.

In this theoretical model (Fig. 1), it is assumed that the Cas9 domain of CBE can bind to ssDNA with a rate $u_0$, initiating the base editing (transition from state 0 to state 2). Alternatively, the protein complex can go to an unproductive state where editing cannot take place, with a rate of $u_4$ (transition from state 0 to state 1). Next, either the Cas9 domain dissociates from DNA with a rate $w_0$ (Transition from state 2 to state 0), or the target cytidine binds to the deaminase catalytic site with a rate $u_1$ (transition from state 2 to state 3). Then the cytidine can either dissociate from the site with a rate $w_1$ without being edited (backward step from state 3 to state 2), or it can be chemically transformed to uridine with a rate $u_3$ (transition from state 3 to state 5). Similarly, the bystander cytidine may bind to the deaminase with a rate $u_2$ (transition from state 2 to state 4), and subsequently, it can either unbind with a rate $w_2$ without being edited (transition from state 4 back to state 2), or it can be chemically transformed with a rate $u_3$ (transition from state 4 to state 6). After that, while Cas9 is still bound to DNA (being in the state 5 or 6), the deaminase can continue editing other cytidines in this region with the same sequence of events (transition to the states 9–12). Alternatively, if Cas9 dissociates from DNA, uridine will be transformed to thymidine through DNA repair (transitions from state 5 to states 7 and 13, or transition from state 6 to states 8 and 14). This U-to-T editing decreases the rebinding rate of Cas9 to ssDNA (transition $7 \rightarrow 5$) if the endogenous DNA repair and replication machinery has changed the DNA sequence from G:C pair to A:T pair. In this case, the new DNA sequence does not perfectly match the spacer sequence of sgRNA. Because the repairing rate is unknown, the rebinding rate is assumed to be $mu_0$ with $0 \le m \le 1$. The parameter $m$ reflects how much the rebinding ability of the BE complex is lowered in comparison with the original substrate. If the DNA repairing rate is slow then $m$ tends to be closer to 1; otherwise, $m$ tends to be closer to 0. Note that the kinetic network in Fig. 1 is a minimal description of complex chemical processes that take place during base editing.

To evaluate the dynamics of base editing, we explored the first-passage probabilities method successfully used in various problems in chemistry, physics, and biology[15–18]. In the case of EGFP site 1 editing by A3A-BE3 there are four possible outcomes as shown in Fig. 1: CTC (state 1, failed editing), CTT (state 13,

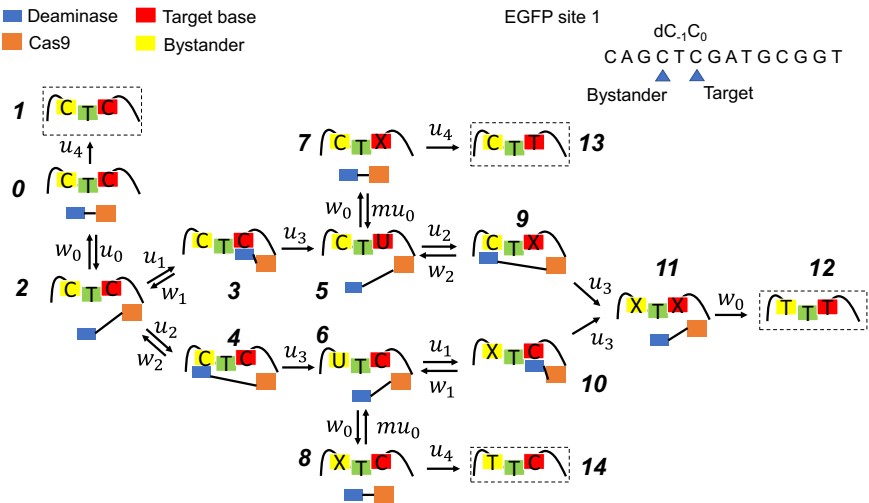

**Fig. 1 Chemical-kinetic model of A3A-BE3 editing the EGFP site 1.** Deaminase, Cas9, target and bystander base are represented by blue, orange, red and yellow squares, respectively. "C" represents cytosine, "T" represents thymine and "X" represents uridine or thymine. Editing is modeled as a multiple-step chemical reaction where Cas9 first binds to ssDNA, cytidine then binds to the catalytic site of the deaminase and is chemically converted to thymidine. Here, $u_i$ and $w_j$ ($j = 0$-4) represent the chemical-kinetic rates for various transitions. The model has a total of 15 states and can produce four outcomes (dashed squares): CTC (failed editing), CTT (only the target base is edited), TTC (only the bystander is edited), TTT (both target and bystander bases are edited).

only the target base is edited), TTC (state 14, only the bystander is edited) and TTT (state 12, both the target base and the bystander are edited). The explicit solution for the probability for the system to end up in one of these products is given below (see derivations in the Supplementary Information Appendix):

$$P_{CTT} = P_5 \cdot \frac{u_4 w_0 (u_3 + w_2)}{(u_2 + w_0)(u_3 + w_2)(u_4 + mu_0) - u_2 w_2 (u_4 + mu_0) - mu_0 w_0 (u_3 + w_2)} \tag{1}$$

$$P_{TTC} = P_6 \cdot \frac{u_4 w_0 (u_3 + w_1)}{(u_1 + w_0)(u_3 + w_1)(u_4 + mu_0) - u_1 w_1 (u_4 + mu_0) - mu_0 w_0 (u_3 + w_1)} \tag{2}$$

$$P_{TTT} = P_5 \cdot \left[ 1 - \frac{u_4 w_0 (u_3 + w_2)}{(u_2 + w_0)(u_3 + w_2)(u_4 + mu_0) - u_2 w_2 (u_4 + mu_0) - mu_0 w_0 (u_3 + w_2)} \right]$$
$$+ P_6 \cdot \left[ 1 - \frac{u_4 w_0 (u_3 + w_1)}{(u_1 + w_0)(u_3 + w_1)(u_4 + mu_0) - u_1 w_1 (u_4 + mu_0) - mu_0 w_0 (u_3 + w_1)} \right] \tag{3}$$

$P_{CTC}$ can be calculated as one minus the sum of the above three probabilities. In equations [1–3], $P_5$ and $P_6$ are two intermediate parameters satisfying:

$$P_5 = \frac{u_0 u_1 u_3 (u_3 + w_2)}{(u_1 + u_2 + w_0)(u_3 + w_1)(u_3 + w_2)(u_0 + u_4) - u_1 w_1 (u_3 + w_2)(u_0 + u_4) - u_2 w_2 (u_3 + w_1)(u_0 + u_4) - u_0 w_0 (u_3 + w_1)(u_3 + w_2)} \tag{4}$$

$$P_6 = \frac{u_0 u_1 u_3 (u_3 + w_1)}{(u_1 + u_2 + w_0)(u_3 + w_1)(u_3 + w_2)(u_0 + u_4) - u_1 w_1 (u_3 + w_2)(u_0 + u_4) - u_2 w_2 (u_3 + w_1)(u_0 + u_4) - u_0 w_0 (u_3 + w_1)(u_3 + w_2)} \tag{5}$$

In experiments, a common way to quantify editing efficiency is to measure the overall probability of editing the target cytidine, $P_t$[11,12]. To compare these predictions with experimental results, $P_t$ was calculated as:

$$P_t = P_{CTT} + P_{TTT} \tag{6}$$

Similarly, the overall probability of editing the bystander cytidine, $P_b$, was calculated as:

$$P_b = P_{TTC} + P_{TTT} \tag{7}$$

Our goal is to parameterize the model by reproducing experimentally measured probabilities, $P_t$ and $P_b$. Here, we assume that the binding between the cytidine (both target and bystander) and the deaminase is mainly a diffusion-controlled process. Therefore, considering that target and bystander cytidine are chemically identical and very close spatially, we added an additional approximation:

$$u_2 = u_1 \tag{8}$$

$$w_2 = w_1 e^{\triangle \triangle E_0 / kT} = w_1 e^{[\triangle E_0 (bystander) - \triangle E_0 (target)] / k_B T} \tag{9}$$

The physical meaning of these expressions is the following: the binding rate to the target or the bystander are the same, but the unbinding is governed by the strength of the interactions between the DNA substrate and the protein complex. In Eq. (9), the term $\triangle E_0$ represents the binding free energy between the ssDNA binding motif and the deaminase. $\triangle\triangle E_0$ represents the difference in $\triangle E_0$ between the dissociation from the target base and the dissociation from the bystander base. This difference arises from the sequence shift in the binding interface. An example is shown in Fig. 1, where the sequence of ssDNA binding motif changes from "$\underline{\mathbf{T}}_{-1}C_0$" in the case of target editing, to "$\underline{\mathbf{G}}_{-1}C_0$" in the case of bystander editing. This change can be formalized by a mutation from thymine to guanine at position -1, which perturbs the binding free energy and further influences the unbinding rate $w$ of the cytidine from the catalytic site. Note that this approximation can also be explained using thermodynamic arguments, since the ratio between rates of binding and unbinding is related to the free energy difference between two states: the state where the protein-RNA complex is bound to the DNA chain and the state where both DNA and protein complex are free, $\frac{u_1}{w_1} = e^{-\triangle E_0(target)/k_BT}$, $\frac{u_2}{w_2} = e^{-\triangle E_0(bystander)/k_BT}$. Using Eq. (8) one can derive the result in Eq. (9).

Similarly, any deaminase mutation can be represented as a perturbation in binding free energy relative to the wild type,

$$w_{1,mutation} = w_{1,WT}e^{\triangle\triangle E_m/k_BT} = w_{1,WT}e^{\left[\triangle E_0(mutation) - \triangle E_0(WT)\right]/k_BT} \tag{10}$$

$\triangle\triangle E_m$ represents the difference in free energy due to mutations.

Substituting Eqs. (8–10) into Eqs. (6–7), we obtain:

estimate $\triangle\triangle E_0$ and $\triangle\triangle E_m$, as shown in the next section. As a result, only two free parameters remain in the model, $\gamma_1$ and $\gamma_3$ (Eqs. 13 and 15), both of which are parameterized by reproducing experimental values of $P_t$ and $P_b$.

**Computational estimates of binding free energy changes.** We chose four CBEs developed by the Joung group:[11] A3A(S99A), A3A(Y130F), A3A(N57Q), and A3A(N57A) to calculate the binding free energy changes between ssDNA and A3A. These CBE variants reduce the bystander effect to different extents while maintaining a high probability of on-target editing. The binding interface in the wild type A3A-ssDNA binding complex is shown in the crystal structure (PDB ID: 5KEG) (Fig. 2a). The carbonyl oxygen of Ser99 forms a hydrogen bond with the N4 atom of the cytidine in the catalytic site ($dC_0$). The hydroxyl group of Tyr130 forms a hydrogen bond with the 5'-phosphate of $dC_0$. Lastly, the nitrogen atom in the sidechain of Asp57 forms a hydrogen bond with the O3 atom of $dC_0$. Therefore, all four CBE variants appear to destabilize the binding between A3A and ssDNA ($\triangle\triangle E_m > 0$) by breaking this hydrogen-bonding network. In addition, since A3A recognizes the $\underline{\mathbf{T}}_{-1}C_0$ instead of the $\underline{\mathbf{G}}_{-1}C_0$ motif, the binding free energy to the deaminase should be higher (more repulsive) for the bystander cytidine than for the target cytidine ($\triangle\triangle E_0 > 0$). To quantitatively calculate $\triangle\triangle E_0$ and $\triangle\triangle E_m$, we utilized the so-called "alchemical free-energy calculations" based on MD simulations[20,21]. A thermodynamic cycle was constructed to convert $\triangle\triangle E_0$ and $\triangle\triangle E_m$ (Fig. 2b, $\triangle G_3 - \triangle G_1$) to the difference between two slow alchemical transitions (Fig. 2b, $\triangle G_2 - \triangle G_4$). One transition is the free energy change for the A3A-ssDNA

$$P_t = \frac{\left(\gamma_1 + m + \gamma_1\gamma_3 + \gamma_1\gamma_2\gamma_3 e^{\frac{\triangle\triangle E_m}{k_BT}}\right)\overline{\left(1 + \gamma_2 e^{\frac{\triangle\triangle E_0 + \triangle\triangle E_m}{k_BT}}\right)} + (\gamma_1 + m)(1 + \gamma_2 e^{\frac{\triangle\triangle E_m}{k_BT}})}{\left(\gamma_1 + m + \gamma_1\gamma_3 + \gamma_1\gamma_2\gamma_3 e^{\frac{\triangle\triangle E_m}{k_BT}}\right) \cdot [(2 + 2\gamma_1 + \gamma_1\gamma_3)\left(1 + \gamma_2 e^{\frac{\triangle\triangle E_m}{k_BT}}\right)\left(1 + \gamma_2 e^{\frac{\triangle\triangle E_0 + \triangle\triangle E_m}{k_BT}}\right) - \gamma_2 e^{\frac{\triangle\triangle E_m}{k_BT}}\left(1 + \gamma_1\right)\left(1 + 2\gamma_2 e^{\frac{\triangle\triangle E_0 + \triangle\triangle E_m}{k_BT}} + e^{\frac{\triangle\triangle E_0}{k_BT}}\right)]} \tag{11}$$

$$P_b = \frac{\left(\gamma_1 + m + \gamma_1\gamma_3 + \gamma_1\gamma_2\gamma_3 e^{\frac{\triangle\triangle E_0 + \triangle\triangle E_m}{k_BT}}\right)\left(1 + \gamma_2 e^{\frac{\triangle\triangle E_m}{k_BT}}\right) + (\gamma_1 + m)(1 + \gamma_2 e^{\frac{\triangle\triangle E_0 + \triangle\triangle E_m}{k_BT}})}{\left(\gamma_1 + m + \gamma_1\gamma_3 + \gamma_1\gamma_2\gamma_3 e^{\frac{\triangle\triangle E_0 + \triangle\triangle E_m}{k_BT}}\right) \cdot [(2 + 2\gamma_1 + \gamma_1\gamma_3)\left(1 + \gamma_2 e^{\frac{\triangle\triangle E_m}{k_BT}}\right)\left(1 + \gamma_2 e^{\frac{\triangle\triangle E_0 + \triangle\triangle E_m}{k_BT}}\right) - \gamma_2 e^{\frac{\triangle\triangle E_m}{k_BT}}\left(1 + \gamma_1\right)\left(1 + 2\gamma_2 e^{\frac{\triangle\triangle E_0 + \triangle\triangle E_m}{k_BT}} + e^{\frac{\triangle\triangle E_0}{k_BT}}\right)]} \tag{12}$$

$$\gamma_1 = u_4/u_0 \tag{13}$$

$$\gamma_2 = w_{1,WT}/u_3 \tag{14}$$

$$\gamma_3 = w_0/u_1 \tag{15}$$

Equations (11–15) give the full analytical expressions in terms of kinetic rates and binding affinities that can be used to calculate the editing probability. There are six free parameters to describe the base editing process ($\gamma_1$, $\gamma_2$, $\gamma_3$, $m$, $\triangle\triangle E_0$, $\triangle\triangle E_m$) but this number can be reduced using additional information. For example, previous binding experiments[19] have indicated that A3A binds to ssDNA with $K_d = 57\mu M$, $K_M = 62\mu M$ and $k_{cat} = 1.1/s$. From these values, one can infer that $w_{1,WT} = 12.54/s$ and $u_3 = 1.1/s$. Therefore, after the cytidine binds to the catalytic site, the relative probability between unbinding and the chemical transformation step, $\gamma_2$, is 11.4. Next, if the changed ssDNA sequence no longer perfectly matches the sgRNA sequence, we assume that successful editing prevents rebinding of Cas9 to ssDNA, therefore $m=0$. Nevertheless, we show below that this assumption only has a minor effect on the final results. Lastly, we performed all-atom computational simulations to

complex due to mutations (Fig. 2b, $\triangle G_2$) whereas the other is the free energy change for A3A alone due to mutations (Fig. 2b, $\triangle G_4$). Calculated values indeed show that mutations cause an apparent increase in the deaminase-ssDNA binding free energy (Fig. 2c), consistent with predictions based on the structural data.

**The rationale for A3A mutants that reduce the bystander effect.** To check whether our model can reproduce the experimentally measured on-target and bystander editing probability, we substituted $\triangle\triangle E_0$ and $\triangle\triangle E_m$ calculated above into Eqs. (11–15) and adjusted $\gamma_1$ and $\gamma_3$. The resulting theoretical prediction is in very good agreement with the experimental measurements[11] (Fig. 3a), with values $\gamma_1 = \frac{u_4}{u_0} = 2.1$ and $\gamma_3 = \frac{w_0}{u_1} = 2.9 * 10^{-5}$. The value of $\gamma_1$ indicates that there is a significant fraction of BEs failing to initiate editing, whereas the value of $\gamma_3$ suggests that the residence time of Cas9 on ssDNA is sufficient for the deaminase to function. We note here that the choice of $m$, which quantifies the effect of sgRNA mismatch on the rebinding rate of Cas9 and ssDNA, does not significantly affect the result (Fig. S1). The model also well produced the editing patterns at multiple genomic loci (Fig. S3), demonstrating the generality of this model.

**a**

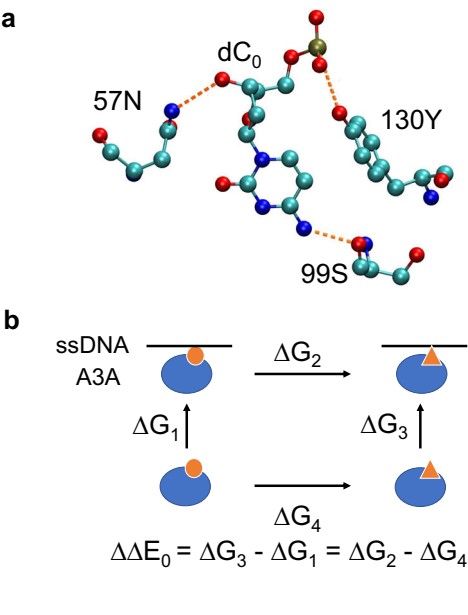

**b**

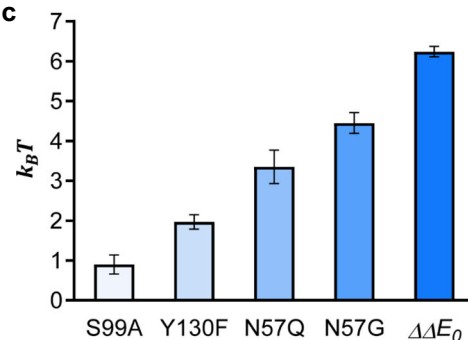

$$\Delta\Delta E_0 = \Delta G_3 - \Delta G_1 = \Delta G_2 - \Delta G_4$$

**c**

**Fig. 2 Calculation of binding free energy changes between ssDNA binding motif and A3A deaminase under various mutations. a** Hydrogen-bonding network between cytidine $dC_0$ and A3A residues 57, 99 and 130. **b** Thermodynamic cycle used to calculate the binding free energy changes: ssDNA (black line), A3A (blue balloons). A mutation at the binding interface is shown by a transition from small orange circle to orange triangle. **c** Computationally estimated changes in binding free energy for A3A deaminase mutations S99A, Y130F, N57Q, N57G and binding free energy change between A3A binding to target and bystander cytidines. The energy unit is $k_BT$ and T = 300 K. Data are presented as mean values ± s.e.m, estimated by Bennett's acceptance ratio method for 200,000 data points. Source data are provided as a Source Data file.

Our theoretical model can be used to explain why the single mutation N57G greatly improves the editing selectivity of A3A-BE3. First, the ratio between the probabilities of having the target cytidine edited before the bystander (Fig. 1, transition state $2 \rightarrow 3 \rightarrow 5 \rightarrow \ldots$ ) and that of the reversed events (Fig. 1, transition state $2 \rightarrow 4 \rightarrow 6 \rightarrow \ldots$ ) can be calculated as:

$$R_1 = \frac{P(state2 \rightarrow 3 \rightarrow \ldots)}{P(state2 \rightarrow 4 \rightarrow \ldots)} = \frac{\frac{u_3}{u_3+w_1}}{\frac{u_3}{u_3+w_2}} = \frac{1 + \gamma_2 e^{\frac{\Delta\Delta E_0+\Delta\Delta E_m}{k_BT}}}{1 + \gamma_2 e^{\frac{\Delta\Delta E_m}{k_BT}}} \quad (16)$$

with $\gamma_2 = 11.4$. In the case of A3A, $R_1$ can be approximated as $e^{\frac{\Delta\Delta E_0}{k_BT}}$. As A3A significantly prefers the TC motif to the GC motif ($\Delta\Delta E_0 \sim 6k_BT$) this ratio is larger than 400. As a result, for both A3A(WT) and A3A(N57G), the probability of having only the bystander edited is very low (Fig. 3b, blue line, almost zero). After the target cytidine is edited (Fig. 1, state 5), the system has the choice of getting released with the product CTT (Fig. 1, state 13)

**a**

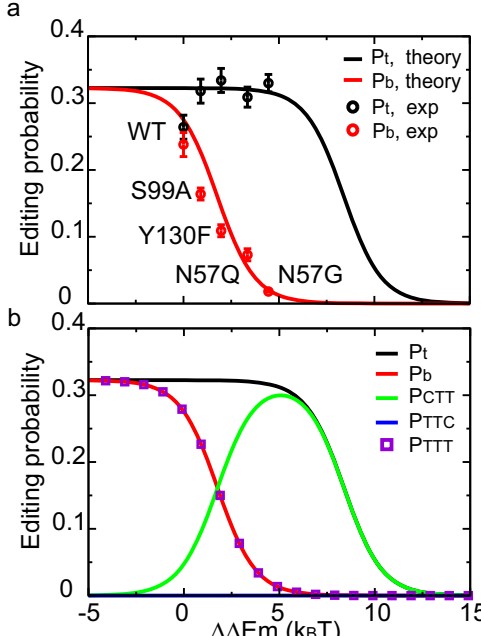

**b**

**Fig. 3 Theoretical model of the A3A-BE3 editing system. a** Comparison between theoretical calculations (solid lines) and experimental data[11] (circles, representing the mean ± s.e.m. of three independent biological replicates). $\Delta\Delta E_m$ represents the perturbation in binding free energy due to the different mutations; $k_BT$ is the unit of energy; $P_t$ and $P_b$ represent the overall probability of editing target and bystander cytidine, respectively. **b** Calculated editing probabilities for products CTT, TTC and TTT. $P_t = P_{CTT} + P_{TTT}$; $P_b = P_{TTC} + P_{TTT}$. Source data are provided as a Source Data file.

or to continue editing the bystander, leading to the product TTT (Fig. 1, state 12). The outcome is largely influenced by the ratio between $w_2$ and $u_3$. If $w_2$ is significantly larger than $u_3$, bystander editing will be blocked because the residence time for the bystander cytidine in the catalytic site is too short to complete the transition to thymidine. Analytically, after the target cytidine gets edited, the probability ratio between CTT and TTT outcomes can be calculated as:

$$R_2 = \frac{P(state5 \rightarrow 13)}{P(state5 \rightarrow 12)} = \frac{w_0}{u_2}\left(1 + \frac{w_2}{u_3}\right) = \gamma_3\left(1 + \gamma_2 e^{\frac{\Delta\Delta E_0+\Delta\Delta E_m}{k_BT}}\right) \quad (17)$$

For A3A(WT), $R_2$ is 0.17, meaning that the dominant product is TTT (Fig. 3b, purple square at $\Delta\Delta E_m = 0$). This explains that for the wild-type A3A the editing efficiency of the target cytidine is similar to that of the bystander. In sharp comparison, as $\Delta\Delta E_m$ increases by 4.5 $k_BT$ for A3A(N57G), $R_2$ is 14.8. Now the dominant edited product changes to CTT (Fig. 3b, green line at $\Delta\Delta E_m =$ 4.5 $k_BT$). In this case, A3A(N57G) minimizes the bystander effect while maintaining a high probability of editing the target base.

Similar arguments can be presented for the other three A3A mutants (S99A, Y130F, and N57Q). Calculations (Table S1) show that $R_2$ gradually increases from 0.42 for S99A to 4.91 for N57Q mutants, indicating that the bystander effect gradually decreases. This is consistent with experimental findings (Fig. 3). However, it is critical to note that to gain high editing selectivity, mutated residues have a non-monotonic effect in the deaminase-ssDNA interface. Here, selectivity is defined as the difference in probabilities between editing the target and editing the bystander. Weakening the binding interface up to 4–6 $k_BT$ (depending on

the system) greatly improves selectivity (Fig. 3a), but it drops when $\triangle\triangle E_m$ continues further increasing. This result can be explained using the following physical considerations. Increasing $\triangle\triangle E_m$ leads to faster-unbinding rates between cytidine and deaminase. At moderate values of $\triangle\triangle E_m$, target editing is less affected (Fig. 1, state 2→5) but bystander editing is blocked (Fig. 1, state 5→12). However, for very large values of $\triangle\triangle E_m$, both editing pathways are essentially blocked and the system prefers to go into the inactive state (Fig. 1, state 1). Therefore, proper modulation of the binding interface is the key to optimize base editing selectivity. We further prove this point in the next section.

**The computational model helps design new A3G-BEs with improved editing selectivity.** In this section, we employ our theoretical model to optimize the editing selectivity of the base editor A3G3.1 (Fig. 4a). First, the editing profiles at both target and bystander bases were calculated by using Eqs. (11–15). Our calculations show that improving the editing selectivity of A3G3.1 requires mutations that increase $\triangle\triangle E_m$ by 2–3 $k_BT$ (shaded area in Fig. 4a). Second, specific mutations were designed and $\triangle\triangle E_m$ was calculated for each mutation by alchemical free-energy calculations as detailed above. Four mutations (T218S, T218N, T218I and T218G) fell into these selection criteria. A failure example is T218W, which loses the editing activity at the target base owing to overly increased $\triangle\triangle E_m$ (Fig. 4a). We then experimentally verified these four mutations at three genomic loci containing the "TCC" motif, including *EMX1* #a3, *PPP1R12C* #a1, and *ATM* #1. We chose these target sites with the "TCC" motif, which are generally more challenging over "ACC" or "GCC" for selectively editing the second C, since "T" and "C" are structurally more similar. In the "TCC" case, the deaminase tends to treat "T" as a "C" and preferentially edits the bystander first "C" as well. In our tests, A3G3.14 (A3G3.1 with T218S) and A3G3.15 (A3G3.1 with T218N) generally show much improved editing selectivity (Fig. 4b), with marginally or modestly decreased editing efficiency. Therefore, A3G3.14 and A3G3.15 were further tested at other five genomic loci, including *MMS22L* #1, *FANCE* #1, *MRPL44* #1, *FANCF* #c1, and *MRPL40* #1 (Fig. 4c). Compared to the original A3G3.1, the target-to-bystander editing ratio increases from average 2.9 to 8.6-fold with mutations.

Our results indicate that mutagenesis stringency and genomic sites are tightly coupled in determining the target-to-bystander editing ratio. Mutagenesis stringency influences the overall editing patterns while specific genomic sites dictate the mutation with the best performance. The basic rule is that relatively large mutagenesis stringency (i.e., high $\triangle\triangle E_m$) is needed for genomic sites with low editing selectivity, and vice versa. Our tested eight genomic loci can be divided into two types in regards to the A3G3.1 editing selectivity. The first group, including *EMX1* #a3, *FANCE* #1, and *MRPL40* #1 sites, showed low selectivity, as expressed by the target-to-bystander editing ratio around 1.07–1.23; whereas the second group, including *PPP1R12C* #a1, *ATM* #1, *MMS22L* #1, *MRPL44* #1 and *FANCF* #c1 sites, showed selectivity to some extent, with the target-to-bystander editing ratio ranging from 2.87 to 5.97. This natural site-dependent difference in selectivity arises from multiple reasons such as sequence context and the levels of DNA accessibility. Therefore, the first group needs mutations with higher $\triangle\triangle E_m$ than the second group. The theoretical model predicts that $\triangle\triangle E_m$ follows an order of S < N (Fig. 4a). As a result, for the first group, A3G3.15 (T218N) generally performs better than A3G3.14 (T218S). In contrast, for the second group, A3G3.14 (T218S) performs better. These results indicate that mutagenesis

stringency and genomic sites should be considered simultaneously during the designing process.

Currently, one difficulty in designing BE is that there are few methods to predict the editing pattern for a novel mutation before experimental validation. In addition, the same mutation can function differently at different genomic loci. Using our model, the editing patterns of those two mutations on A3G3.1 are computationally predictable, and well-validated by experiments (Fig. 4b and 4c). This result demonstrates the power of combining theoretical and experimental approaches. *EMX1* #a3 site was also tested in three cell lines, K562, Jurkat, and HeLa (Supplementary Fig. 4). Although these cell lines generally have low transfection efficiency, we still observed an increase of the target-to-bystander editing ratio in A3G3.15 treated cells, compared to those treated by A3G3.1 (Supplementary Fig. 4 and Fig. 4b), for two cell lines, K562 (two-tailed $p = 0.0005$ with unpaired $t$-test) and Jurkat ($p = 0.0001$). The improvement for Hela cells is less significant and needs further optimization in the future.

**Discussion**
In this work, we developed a theoretical framework to understand the molecular mechanisms of base editing. Our approach suggests several general rules to design BEs with improved editing selectivity. This goal is fulfilled by modulating the binding affinity between deaminase and ssDNA using mutagenesis ($\triangle\triangle E_m$). The principle is to guarantee that the residence time of deaminase on ssDNA is sufficiently long to complete the editing of the first on-target site, while being too short for editing the second (bystander) site. Our theoretical method predicts optimal values for $\triangle\triangle E_m$. Away from these optimal values, selectivity decreases. Therefore, instead of testing experimentally a set of candidate BE mutants, one can instead set up a computational pre-screening process by estimating the $\triangle\triangle E_m$ of those variants, and only candidates near the optimal value can then be tested experimentally. Herein, we used alchemical free-energy calculations to estimate $\triangle\triangle E_m$. The accuracy of this method has been validated in the A3A and A3G system (Figs. 3 and 4). Future work will help to develop carefully parameterized scoring functions for ssDNA-protein interactions or combine machine learning techniques[22] so that the prediction of $\triangle\triangle E_m$ can be accelerated. In addition, when estimating $\triangle\triangle E_0$ and $\triangle\triangle E_m$, our model only considers the local sequence near the target base and neglects the long-range contributions from other bases in the same editing window. Because the local sequence context is a major chemical factor in determining the relative outcomes of bystander edits vs target site edits, with such approximations our model can still explain the existing experimental data (Fig. 3) and guide the design of new mutations (Fig. 4). However, the long-range contributions might serve as additional regulators and requires more detailed investigations in the future.

Equations (11–15) indicate that for a given system the editing probability is regulated by two other parameters, $\gamma_1$ and $\gamma_3$, in addition to $\triangle\triangle E_m$. Therefore, we plotted the editing probability for different values of $\gamma_1$ (Fig. 5a) and $\gamma_3$ (Fig. 5b). We first reduced the parameter $\gamma_1$ (Fig. 5a) which can be achieved by increasing the on-rate of Cas9 to ssDNA. It turns out that the editing selectivity for the WT system is not affected by $\gamma_1$ (Fig. 5a, solid blue line vs dashed blue line at $\triangle\triangle E_m = 0$) as the efficiencies of both target and bystander editing increase synchronously. However, the selectivity greatly improves when $\triangle\triangle E_m$ is 4–6 $k_BT$, meaning that $\gamma_1$ amplifies the deaminase mutation regulation effect. This suggests an effective combination strategy in the design of highly selective BE: optimization of $\triangle\triangle E_m$ first, then reducing $\gamma_1$ to amplify this effect. We then reduced the parameter $\gamma_3$ (Fig. 5b). Our calculation indicates that reducing $\gamma_3$ does not change the maximum editing selectivity but induces a

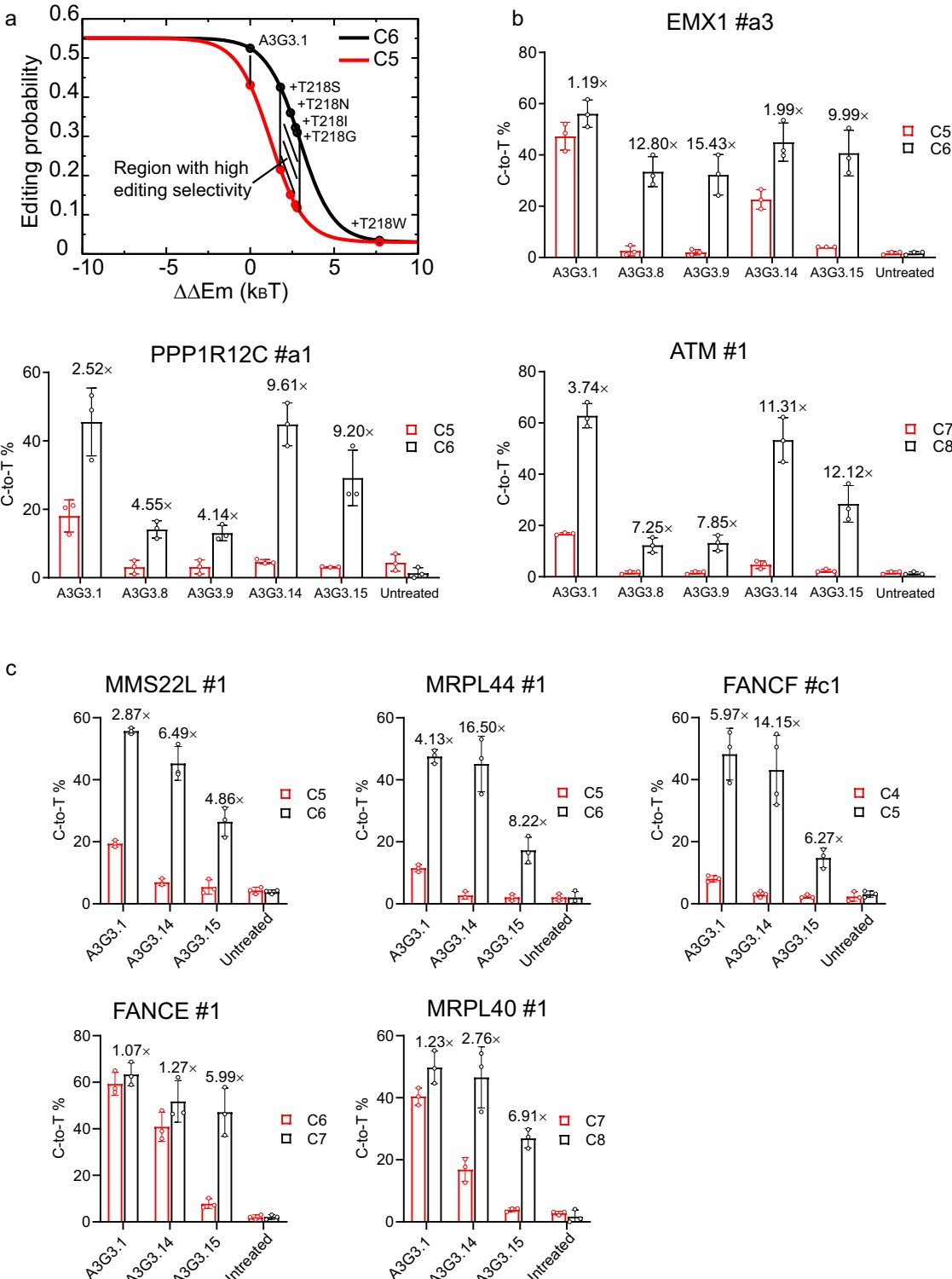

**Fig. 4 Engineering of A3G-BEs. a** Theoretical calculation. C6 represents the target base. C5 represents the bystander. The shaded area represents the region with improved editing selectivity. **b** Experimental measurements at three genomic loci for four mutations picked by theoretical model; A3G3.1 is the full-length APOBEC3G deaminase with a set of mutations which increase the catalytic efficiency. A3G3.8, 3.9, 3.14, and 3.15 are A3G3.1 with T218G, T218I, T218S, and T218N, respectively. Bar plots represent the mean ± s.d. of three independent biological replicates. **c** Experimental measurements at other five genomic loci for A3G3.14 (T218N) and A3G3.15 (T218S). Bar plots represent the mean ± s.d. of three independent biological replicates, except for the bar representing the editing efficiency of A3G3.14 at FANCF #c1 site, which shown the mean ± s.d. from four biological replicates. Source data are provided as a Source Data file.

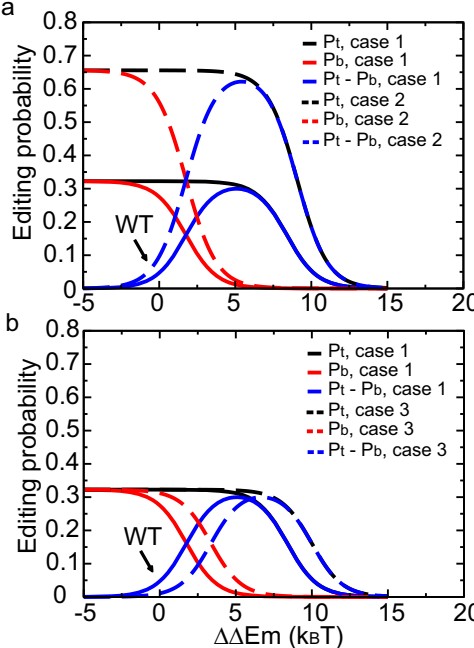

**Fig. 5 Theoretical calculations on regulating the base editing pattern of A3A-BE3.** Base editing pattern of A3A-BE3 regulated by (**a**) $\gamma_1$ and (**b**) $\gamma_3$. The definition of $\triangle\triangle E_m$, $\gamma_1$ and $\gamma_3$ can be found in Eqs. ([10,13–15]). $P_t$ and $P_b$ are the overall probabilities of editing the target and bystander cytidine, respectively. The difference between $P_t$ and $P_b$ is shown in blue. The setting with original parameters is represented by solid lines (case 1) whereas variants are represented by dashed lines (case 2: $\gamma_1$ divided by five; case 3: $\gamma_3$ divided by five). Source data are provided as a Source Data file.

right shift in the editing profile, i.e., a larger $\triangle\triangle E_m$ value is required to achieve the maximum editing selectivity.

Another interesting question is whether a general mutation to all BEs homologs exists that optimizes editing. Unfortunately, we found that a mutation working perfectly for one type of deaminase may fail for another type, even when they are homologs. For example, A3A mutations N57G and Y315 greatly reduce the bystander effect while maintaining a high probability of target editing, but A3G N244G almost loses the base editing ability (A3A N57 aligns with A3G N244) (Supplementary Fig. 2a). The theoretical model developed above explains this negative outcome, since $\triangle\triangle E_m$ for N244G turns out to be much larger than its optimal value (Supplementary Fig. 2b). Our model shows that an energy shift of only few $k_BT$ can dramatically change the selectivity of a BE, from its optimal value to zero. This is due to subtle differences between homologs: the on-rate of A3G binding to ssDNA is lower that for A3A, as the former can form a large dimer that interferes with binding. In addition, A3G has lower $\triangle\triangle E_0$ (see Eq. ([9])) because the sequence of the ssDNA binding motif changes from "$\underline{C_{-1}}C_0$" in the target editing to "$\underline{T_{-1}}C_0$" in the bystander editing. The energy perturbation from $\underline{C_{-1}}$ to $\underline{T_{-1}}$ is smaller than that from $\underline{T_{-1}}$ to $\underline{G_{-1}}$ in the case of A3A. These differences cause A3A and A3G to have distinct behavior under equivalent mutations. In fact, even for the same BE, editing different loci can change the binding free energy by a few $k_BT$ because at different loci the neighboring bases of the bystander may vary. Therefore, each BE may require a unique optimization to achieve high editing selectivity. Nevertheless, our approach gives quantifiable parameters that can be used to accelerate the search for best editors. In summary, a general design strategy would be (a) employing the chemical-kinetic model (Eqs. ([11–15])) to determine the binding free energy changes required

to achieve the maximum editing selectivity, $\triangle E_{peak}$; (b) designing mutations in the deaminase, estimating $\triangle\triangle E_m$ and selecting the ones near $\triangle E_{peak}$; (c) keeping mutations picked in the previous step and designing extra mutations that increase the binding rate of Cas9 to DNA substrate; (d) experimental validation of these changes.

## Methods

**Free energy calculations by molecular dynamic simulations.** We utilized MDs based on alchemical free-energy calculations[20,21] (Fig. [2]) to estimate the binding free energy changes under various mutations[11]. All simulations were carried out using the Gromacs package[23]. Amber99sb*ILDN force field with bsc0 correction for nucleic acids was used[24–27]. The integration time step was set to 2 fs. The initial states of the A3A-ssDNA[28] and the A3G-ssDNA[29] binding complex were taken from their crystal structures (PDB ID: 5keg and 6bux, respectively). Then we used the pmx webserver[30,31] to generate hybrid structures and topologies representing mutations. Each system was solvated in a cubic box with TIP3P water molecules. The ions concentration was set to 0.1 M. The dimension of the box is 9 nm. The temperature was maintained at 300 K by the Berendsen thermostat[32] while the pressure was maintained at 1.0 atm by using the Parrinello-Rahman barostat[33]. Electrostatic interactions were calculated by the Particle Mesh Ewald method[34]. The soft-core function was used for the nonbonded interactions during the alchemical transitions[35]. For each system, energy minimization was first performed, followed by 1 ns NVT and 1 ns NPT equilibration with the protein configuration restrained. Then the system was further equilibrated for 5 ns without any restrain. The last snapshot of the trajectory served as the starting configuration for the following alchemical transitions.

The alchemical transition ($\lambda = 0 \rightarrow \lambda = 1$) was divided into 21 consecutive windows with the bin size of 0.05. For each window $i$, $\lambda$ was first increased from 0 to $\lambda_i (\lambda_i = 0, 0.05, 0.1, 0.15\ldots)$ with a slow rate $10^{-8}$/step, then was fixed to $\lambda_i$ for 40 ns production run. dH/dl values were recorded every 100 steps. The free energy and error bar were estimated by Bennett's acceptance ratio method[36].

## Experiment

*Mammalian cell culture.* HEK293T cells (American Type Culture Collection, CRL-3216) were cultured in GlutaMAX^TM high-glucose Dulbecco's modified Eagle's medium (DMEM, Thermo, cat. 10569044). HeLa (ATCC, CCL-2), K562 (ATCC, CRL-3343), and Jurkat (ATCC, TIB-152) lines were maintained in GlutaMAX^TM RPMI 1640 medium (HEPES buffered, Thermo, cat. 72400146). The culture media were all supplemented with 10% fetal bovine serum (FBS, Thermo, cat. 10437028) and 100 U/mL penicillin-streptomycin (Thermo, cat. 15140122). Cells were grown in a humid atmosphere at 37 °C with 5% CO₂. HEK293T and HeLa cells were passaged at a ratio of 1:4 when reaching 90% confluency by using TrypLE Express (Thermo, cat. 12605028). Jurkat and K562 cells were subcultured and added with fresh medium every 2 or 3 days to keep the density below $10^6$ cells/mL. Mycoplasma testing was performed monthly using a mycoplasma PCR detection kit (abm, cat. G238).

*Plasmid construction.* The full-length human codon-optimized wild-type A3G with a set of mutations (P200A + N236A + P247K + Q318K + Q322K) was synthesized as gBlock and inserted into the BE4max construct (Addgene #112093) to replace the rAPOBEC1 region, resulting in A3G3.1. To do so, both insertion and vectors were amplified using primers with overhangs containing Esp3I recognition sites, which would generate compatible complementary sticky ends after cutting. Then the one-pot Golden Gate assembly was employed to cut and ligate two amplified pieces by using Esp3I and T4 DNA ligase (New England Biolabs, cat. R0734L and M0202L). Likewise, the Y315F and N244G variants were respectively generated by designing an extra pair of primers containing the indicated mutations and performing a three-piece assembly. The A3G3.8, 3.9, 3.14, and 3.15 constructs were Golden Gate cloned to introduce point mutations T218G, T218I, T218S, and T218N to A3G3.1.

**Cell transfection, genomic DNA extraction, amplicon sequencing, and analysis.** Cell transfection was performed as previously described with slight modifications[12]. Briefly, HEK293T or HeLa cells were seeded into a poly-D-lysine–coated 48-well plate (Corning, cat. 354509) at a density of $4.5 \times 10^4$ cells per well in 250 µL antibiotic-free culture medium supplemented with 10% FBS. In about 12–16 h, upon reaching 70% confluency, cells of each well were transfected. K562 and Jurkat cells were reverse-transfected at a density of $2 \times 10^4$ cells per well. Four cell types were all transfected with 750 ng BE plasmids and 250 ng sgRNA plasmids using 1.5 µl Lipofectamine 2000 (Thermo, cat. 11668019) dispersed in 25 µL Opti-MEM (Thermo, cat. 31985062) according to the manufacturer's instructions. Three days later, the genomic DNAs were collected by removing the medium by aspiration or centrifugation, washing the cells gently with PBS (Thermo, cat. 10010049), and lysing the cells at 37 °C for 1–2 h with 100 µl per well of lysis buffer containing 10 mM Tris-HCl (pH 7.5, Thermo, cat. 15567027), 0.05% SDS (Sigma, cat. 71725), and 25 µg/mL proteinase K (Fisher BioReagents, cat. BP1700). The cell lysates containing genomic DNA were then subjected to heat

inactivation of the proteinase K at 80 °C for 0.5–1 h. For Sanger sequencing, the genomic DNA amplification primers and the Sanger sequencing primers are listed in Supplementary Table 2. The 20 µL PCR reactions containing 0.4 U Q5 High-Fidelity DNA polymerase (New England Biolabs, cat. M0491L), 0.5 µM of forward and reverse primers, and 100 ng of genomic DNA, were performed using a 35-cycle PCR program. The Sanger sequencing results were analyzed using EditR online software (http://baseeditr.com/).

A total of 100 ng genomic DNA was amplified at the *EMX1* target site by using primers attached with the partial Illumina adapters and 8 bp compatible and nucleotide-balanced indices on both 5′ and 3′ end. The forward and reverse primers are as follows. For: 5′-ACACTCTTTCCCTACACGACGCTCTTCCGAT CTNNNNNNNNTGTGGTTCCAGAACCGGAG-3′; Rev: 5′-GACTGGAGTTCA GACGTGTGCTCTTCCGATCTNNNNNNNNCTCTGCCCTCGTGGGTTT-3′. The protospacer sequence is 5′-GAGTCCGAGCAGAAGAAGAA-3′. The amplicon sequence is 5′-TGTGGTTCCAGAACCGGAGGACAAAGTACAAA CGGCAGAAGCTGGAGGAGGAAGGGCCTGAGTCCGAGCAGAAGAAGAAG GGCTCCCATCACATCAACCGGTGGCGCATTGCCACGAAGCAGGCCAATG GGGAGGACATCGATGTCACCTCCAATGACTAGGGTGGGCAACCACAAA CCCACGAGGGCAGAG-3′.

Amplicons were pooled, column purified (Qiagen), recovered in nuclease-free water (Thermo, cat. 10977023), and quantified by the Qubit dsDNA HS assay (Thermo, cat. Q32851). A volume of 25 µL sample with the final concentration adjusted to 20 ng/µL was submitted for Amplicon-EZ sequencing (Genewiz). Fastq files were then downloaded from Genewiz Ftp server and analyzed by using CRISPResso2 (https://github.com/pinellolab/CRISPResso2) to align reads and quantify the base editing efficiency and frequency[37].

All gblocks and primers were synthesized by Integrated DNA technologies.

**Statistical analysis**. All experiments were performed with 2–4 independent biological replicates. Bar plots in Fig. 4, and Supplementary Fig. 4 represent means ± standard derivation (s.d.). Bar plot in Supplementary Fig. 2a represents means ± standard error of the mean (s.e.m.).

**Reporting summary**. Further information on research design is available in the Nature Research Reporting Summary linked to this article.

## Data availability
All data generated or analyzed during this study are included in this published article (and its supplementary information files). Plasmids encoding A3G CBEs used in this study will be made available upon reasonable request to the corresponding authors. Targeted amplicon sequencing data have been deposited at the Sequence Read Archive (SRA): https://www.ncbi.nlm.nih.gov/bioproject/PRJNA700693. Source data are provided with this paper.

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

## Acknowledgements
Q.W. acknowledges the funding support from "USTC Research Funds of the Double First-Class Initiative" (YD2030002006), the National Natural Science Foundation of

China (32000882), and from the Center for Theoretical Biological Physics sponsored by the NSF (PHY-2019745). A.B.K. acknowledges the support from the Welch Foundation (C-1559), and from the NSF (CHE-1953453 and MCB-1941106); X.G. acknowledges the funding support from the NIH (R01HL157714) and the Rice University Creative Ventures Fund. The numerical calculations in this paper have been done on the supercomputing system in the Supercomputing Center of the University of Science and Technology of China and the Supercomputing Center of Rice University.

## Author contributions

Q.W., X.G., and A.B.K. designed the research; Q.W. and Z.C.Z. performed the theoretical study; J.Y. and J.A.V. performed the experimental study; Q.W., J.Y., X.G., and A.B.K. analyzed the data; Q.W., J.Y., X.G., and A.B.K. wrote the paper.

## Competing interests

The authors declare no competing interests.
