## [Peer Review File · Nature Communications]

Reviewers' Comments:

Reviewer #1:

Remarks to the Author:

In this paper, Wang and colleagues focus on base editors to understand the basis of discrimination between target and off-target bases, a severe issue for the application of base editing. They employ a statistical approach to seek for a general theoretical framework to guide the design of mutations that can lead to high editing activity at the target base and low activity at bystander base. The theoretical approach is interesting and potentially valuable as a predictive tool. The paper, however, is extremely technical and very hard to follow. It is not clear how the experimental community could rely on this very complicated model to predict base discrimination. This is the main reason why it is not appropriate for the readership of Nature Communications, which is very broad and will not understand the theoretical framework as described by the authors. This paper is appropriate for a specialized theoretical journal.

There are issues with the molecular dynamics simulations. The authors use a force field that is not appropriate for nucleic acids and has shown to report inconsistencies for single stranded DNA, which is the case reported here. They should employ the parameters with the bsc0 corrections for nucleic acids (Perez et al., 2007). This force field corrects the α/γ transitions and avoids unphysical transitions. For RNA, the corrections were added to balance the anti and the high-anti conformations of the χ angle (Banas et al., 2010; Zgarbova et al., 2011). The dynamics of nucleic acids is tightly dependent on metal ions concentrations, but it is not clear at which ionic concentration the simulations have been performed. These issues are fixable, but paper remains not appropriate for Nature Communication for the reasons reported above.

Reviewer #2:

Remarks to the Author:

Summary

The authors present a well-written and well-explained computational strategy for designing base editors with reduced or eliminated bystander editing frequencies. They use previously published mutations that have been shown to reduce bystander effects in the highly active deaminase A3A as their test enzyme and used their findings to rationalize the editing frequencies of a set of previously published A3G-based cytosine base editor mutants.

Major comments

The assumption that uridine is "quickly" transformed to thymidine through DNA repair is troubling (page 4). DNA repair velocity will be highly dependent on cell type, and has not been measured precisely in the literature as far as the references show. The authors should explain or qualify this assumption further. Similarly, the idea that re-binding is slowed after editing does not refer to the fact that for this to be the case not only does deamination have to occur but correction of the opposite strand (the sgRNA target strand) must also be repaired through DNA repair or replication.

The authors give a very good description of how $\Delta\Delta E0$ relies on the motif in which the base is located, focusing on the example of EGFP in which the target base is in a TC motif whereas the bystander base is in a GC motif. While the local sequence context is indeed a major factor in determining the relative outcomes of bystander edits vs. target site edits, the contribution of position of the target vs. the bystander within the editing window has not been addressed. Since the relative position of a base within the editing window of 4-10 is an additional key factor determining the preference of the base editor for a target base over a bystander base, this must be considered explicitly.

The data in Figure 2 is extremely interesting and it is great to see how closely the predictions match the measured target and bystander editing frequencies. However, in order to make conclusions about the generality of this tool, the authors must include at a minimum 6 different target sites for which the same calculations and experiments have been performed. Ideally the authors would also perform these experiments in at least three different cell types.

In the final part of this work, the authors applied their computational framework to some previously published A3G mutants with the goal of increasing target editing while reducing bystander editing. Although statistically significant, the improvement in target editing is extremely modest and only measured at one target site. Even more worryingly, in their “improved” mutant A3G(B+C) also has increased bystander editing which appears to be proportional to the increase in target site editing.

It is disappointing that the authors do not use their computational platform to design (or guide design of) any novel mutations. Instead, they use their framework to explain the behavior of previously engineered deaminase mutations. This is a critical limitation of this work and the authors must show that they can use their modeling platform to design new mutations that will reduce bystander editing while increasing or maintaining target editing at multiple genomic loci, ideally in multiple cell types.

Minor comments

The authors refer to adenine base editors as adenosine base editors. Although this can be quite a misleading point due to the fact that the base editors actually act on neither adenine nor adenosine, for consistency with the field they should refer to them as adenine base editors. In addition, the authors correctly refer to cytosine base editors as such, but the class of deaminase employed in cytosine base editors is referred to as the “cytidine deaminase” family, not “cytosine deaminase” as the authors write in the introduction.

The target site protospacer in Fig 1 is very confusing – the sequences should not be offset.

The description of the non-complementary effects on bystander base editing of a) weakening the binding interface (reduces bystander editing) and b) increasing $\Delta\Delta E_m$ on page 7 is effective, clear and concise. The authors should show (perhaps in the SI) their calculations for the other mutants.

The methods describing amplification of their gDNA is not sufficient for reproduction; key details such as the polymerase used and cycle number are missing. Further, the type of replicates has not been addressed (biological vs. technical), and for the experimental data in Figure 2 there are no replicates shown.

Reviewer #1

We thank Reviewer 1 for the helpful suggestions to our manuscript.

1. >> In this paper, Wang and colleagues focus on base editors to understand the basis of discrimination between target and off-target bases, a severe issue for the application of base editing. They employ a statistical approach to seek for a general theoretical framework to guide the design of mutations that can lead to high editing activity at the target base and low activity at bystander base. The theoretical approach is interesting and potentially valuable as a predictive tool. The paper, however, is extremely technical and very hard to follow. It is not clear how the experimental community could rely on this very complicated model to predict base discrimination. This is the main reason why it is not appropriate for the readership of Nature Communications, which is very broad and will not understand the theoretical framework as described by the authors. This paper is appropriate for a specialized theoretical journal.

We thank the reviewer for this comment. In the revised manuscript, to let readers better understand how our work helps the experimental community, we added a new section “**The computational model helps design BE with improved editing selectivity**” in Page 8 to elaborate the process of using our computational model to guide the experimental design.

In addition, more explanations of the model and the details of calculations are added in the text. We should also mention that our chemical kinetic model provides a minimal description. At the same time, although the model has many states, the calculations are very straightforward and they have clear physical meaning. This is why our theoretical model is powerful and appealing.

2. >> There are issues with the molecular dynamics simulations. The authors use a force field that is not appropriate for nucleic acids and has shown to report inconsistencies for single stranded DNA, which is the case reported here. They should employ the parameters with the bsc0 corrections for nucleic acids (Perez et al., 2007). This force field corrects the α/γ transitions and avoids unphysical transitions. For RNA, the corrections were added to balance the anti and the high-anti conformations of the χ angle (Banas et al., 2010; Zgarbova et al., 2011). The dynamics of nucleic acids is tightly dependent on metal ions concentrations, but it is not clear at which ionic concentration the simulations have been performed. These issues are fixable, but paper remains not appropriate for Nature Communication for the reasons reported above.

We thank the reviewer for this important comment. In the revised manuscript, all simulations were re-performed by using the force field with the bsc0 corrections. The free energies can show 1-2 $k_B T$ differences compared to previous calculations.

However, the main results qualitatively and even quantitatively in most cases remain the same (Fig. 2&3). This suggests that our theoretical approach probably captures the correct physics of the process of base editing. The ionic concentration was set to 0.1M. This information is now added in the Methods section.

Reviewer #2

We thank the Reviewer 2 for the positive comments and constructive suggestions to our manuscript.

Major comments

1. >> The assumption that uridine is “quickly” transformed to thymidine through DNA repair is troubling (page 4). DNA repair velocity will be highly dependent on cell type, and has not been measured precisely in the literature as far as the references show. The authors should explain or qualify this assumption further. Similarly, the idea that re-binding is slowed after editing does not refer to the fact that for this to be the case not only does deamination have to occur but correction of the opposite strand (the sgRNA target strand) must also be repaired through DNA repair or replication.

Thanks for your comments and we provide more details of our model in the revised draft. First, the result of our kinetic model does not rely on how fast uridine is transformed to thymidine. If this transformation is fast, then state 7 (CTX) in Fig. 1 is CTT. Otherwise, state 7 is CTU and it will be transformed to CTT later with a certain rate u' . However, in our kinetic model, the probability of the final product depends only on the choice at the branched point, i.e., the relative probability of transition $5 \rightarrow 7$ versus $5 \rightarrow 9$. Any event after reaching state 7 will not influence the “probability” of the outcome.

Second, it is true that the re-binding rate of deaminase is determined by how fast the repairing finishes. As the reviewer pointed out, the repairing rate is unknown. That is why we used a parameter m in the model to control the re-binding rate. The case $m=0$ corresponds to the situation when the repairing is fast, while $m=1$ indicates that the repairing is slow. However, we found that the choice of m does not influence the results significantly (see Fig. S1), implying that m is not the key parameter to regulate the editing selectivity. For this reason, we chose in our calculations $m=0$ to simplify the analysis.

We rewrote the modeling part to explain these arguments better in Page 4.

2. >> The authors give a very good description of how $\Delta\Delta E_0$ relies on the motif in which the base is located, focusing on the example of EGFP in which the target base is in a TC motif whereas the bystander base is in a GC motif. While the local sequence context is indeed a major factor in determining the relative outcomes of bystander edits vs. target site edits, the contribution of position of the target vs. the bystander within the editing window has not been addressed. Since the relative position of a base within the editing window of 4-10 is an additional key factor determining the preference of the base editor for a target base over a bystander base, this must be considered explicitly.

We thank the reviewer for pointing this out. Indeed, when estimating $\Delta\Delta E_0$, we only

consider the “local” sequence adjacent to the target base and the bystander base. The long-range contributions from other bases in the same editing window are neglected for now. This limitation is rather technical because the free energy calculations through molecular dynamic simulations rely on the crystal structures. Unfortunately, the full structural information is only available for bases adjacent to the target base in the crystal structure of the A3A-DNA binding complex (PDB ID: 5keg). Therefore, we apologize that we cannot easily include the long-range contributions in the current framework. However, we emphasize the following two points:

First, the local bases are the most important factors to influence the binding affinity between DNA and deaminase. Regardless of the nearest-neighboring approximation, our method can still explain the existing experimental data (Fig. 3) and guide the design of new mutations (new Fig. 4). Second, in the theoretical model, i.e., eqns. [11-15], $\Delta\Delta E_0$ represents the real binding free energy changes between target editing and bystander editing with all bases counted. In the future, $\Delta\Delta E_0$ can be obtained from binding experiments, or fitted as a free parameter if sufficient mutation data exist. Under those circumstances, eqns. [11-15] still hold and can provide better predictions.

We addressed the limitation of our model in the discussion section (Page 9). The model will be extended in the future.

3. >> The data in Figure 3 is extremely interesting and it is great to see how closely the predictions match the measured target and bystander editing frequencies. However, in order to make conclusions about the generality of this tool, the authors must include at a minimum 6 different target sites for which the same calculations and experiments have been performed. Ideally the authors would also perform these experiments in at least three different cell types.

We applied our model to six different target sites edited by A3A-BE, as requested by the reviewer (Fig. S3). Two cell types, human HEK293T and erythroid precursor cell were included in the data. It is shown now that the calculations reproduce the editing patterns of A3A observed in experiments. [Note: the experimental data comes from other labs in the literature (Gehrke JM, et al. (2018) An APOBEC3A-Cas9 base editor with minimized bystander and off-target activities. Nat. Biotechnol. 36(10):977-982, Fig. 1, Fig. S1 and Fig. S9)]

We also applied our model to design new mutations in A3G-BEs and tested at multiple genomic loci and in multiple cell types. Please see the response to question (5).

Figure S3. Theoretical model reproducing the editing patterns of six genomic loci observed in experiments (ref [11]). The cell type is human HEK293T except the last one is erythroid precursor cell.

4. >> *In the final part of this work, the authors applied their computational framework to some previously published A3G mutants with the goal of increasing target editing while reducing bystander editing. Although statistically significant, the improvement in target editing is extremely modest and only measured at one target site. Even more worryingly, in their “improved” mutant A3G(B+C) also has increased bystander editing which appears to be proportional to the increase in target site editing.*

We agree that this is not the best example to show the ability of our model. It is because in this case, the editing selectivity of the wide-type BE is already high, leaving little room to be optimized. Therefore, in the revised manuscript, we optimized another base editor with low editing selectivity in the wide-type. We designed novel mutations as requested and tested at multiple genomic loci as well as in multiple cell types. Please see the response to question (5).

5. >> *It is disappointing that the authors do not use their computational platform to design (or guide design of) any novel mutations. Instead, they use their framework to explain the behavior of previously engineered deaminase mutations. This is a critical limitation of this work and the authors must show that they can use their modeling platform to design new mutations that will reduce bystander editing while increasing or maintaining target editing at multiple genomic loci, ideally in multiple cell types.*

We thank the reviewer for pointing this out. In the revised manuscript, we successfully designed two novel mutations T218S and T218N that have not been proposed for A3G BE in the literature. The mutations were tested at eight genomic sites in HEK293T cells (Fig. 4): EMX1 #a3, PPP1R12C #a1, ATM #1, MMS22L #1, FANCE #1, MRPL44 #1, FANCF #c1, and MRPL40 #1. Compared to the original A3G3.1, the target-to-bystander editing ratio increases from average 2.9 to 8.6-fold with mutations. Then EMX1 #a3 site was tested in three cell lines, K562, Jurkat, and HeLa (Fig. S4). Although these cell lines generally have low transfection efficiency, we still observed an increase of the target-to-bystander editing ratio in A3G3.15 treated cells, compared to those treated by A3G3.1. This result demonstrates the power of combining theoretical and experimental approaches. Please see details in the new section **“The computational model helps design BE with improved editing selectivity”** in Page 8.

Figure 4. Engineering of A3G-BEs. (A) Theoretical calculations. C6 represents the target base. C5 represents the bystander. Shaded area represents the region with improved editing selectivity; (B) Experimental measurements at three genomic loci for four mutations picked by theoretical model; A3G3.1 is the full-length APOBEC3G deaminase with “set A” mutations which increase the catalytic efficiency. A3G3.8, 3.9, 3.14 and 3.15 are A3G3.1 with T218G, T218I, T218S, and T218N, respectively. (C) Experimental measurements at other five genomic loci for A3G3.14(T218N) and

A3G3.15(T218S). For (B) and (C), bar plots represent mean \pm s.d. of three or four independent biological replicates.

Figure S4. Experimental measurements at EMX1 #a3 site for A3G3.1, A3G3.14 (T218S) and A3G3.15 (T218N) using three cell lines, K562 (A), Jurkat (B), and HeLa (C). Bar plots represent mean \pm s.d. of two or three independent biological replicates.

Minor comments

>> *The authors refer to adenine base editors as adenosine base editors. Although this can be quite a misleading point due to the fact that the base editors actually act on neither adenine nor adenosine, for consistency with the field they should refer to them as adenine base editors.*

In addition, the authors correctly refer to cytosine base editors as such, but the class of deaminase employed in cytosine base editors is referred to as the “cytidine deaminase” family, not “cytosine deaminase” as the authors write in the introduction.

We apologized for the confusion. Now those misprints are corrected.

>> *The target site protospacer in Fig 1 is very confusing – the sequences should not be offset.*

This problem is fixed.

>> *The description of the non-complementary effects on bystander base editing of a) weakening the binding interface (reduces for bystander editing) and b) increasing $\Delta\Delta E_m$ on page 7 is effective, clear and concise. The authors should show (perhaps in the SI) their calculations for the other mutants.*

We added Table S1 in the SI to show calculations for all mutations in Fig. 3A.

>> *The methods describing amplification of their gDNA is not sufficient for reproduction; key details such as the polymerase used and cycle number are missing. Further, the type of replicates has not been addressed (biological vs. technical), and for the experimental data in Figure 2 there are no replicates shown.*

We added details to the Methods section, and Figure 4, S2 and S4 legends. The primer sequences were shown as Supplementary Table S2. Experimental data in all figures were represented with error bars from independent biological replicates.

Reviewers' Comments:

Reviewer #1:

Remarks to the Author:

The authors have addressed the technical concerns. However, the paper remains very technical and not appropriate for the audience of Nat. Commun. The wording, figures and the overall text makes it appropriate for a specialized journal in the field of Physical Chemistry. If published, the interdisciplinary readership of Nat. Commun. will find hard time understanding this paper.

Reviewer #2:

Remarks to the Author:

The authors have, for the most part, addressed reviewer comments. The manuscript is much clearer now and should be more accessible for a broader audience. Thanks!

One concern has arisen from the new Fig 4 and corresponding SI Figure. An unspecified "control" is shown for each target site - this needs to be clearly defined. If "control" is untreated cells the authors should describe this. If so, it is surprising that some non-zero level of editing is observed in the control.

Further, the data in SI Fig 4 is not clear that there is a significant improvement of editing selectivity (at least while maintaining on-target editing efficiency). This either needs to be addressed with a statistical test or the text should be updated to reflect this.

Reviewer #1

The authors have addressed the technical concerns. However, the paper remains very technical and not appropriate for the audience of Nat. Commun. The wording, figures and the overall text makes it appropriate for a specialized journal in the field of Physical Chemistry. If published, the interdisciplinary readership of Nat. Commun. will find hard time understanding this paper.

We respectfully disagree with the opinion of the reviewer. All results presented in our work are explained using simple physical arguments. All the mathematical calculations are done at the level of undergraduate Chemistry or Biology class. Before submitting, we shared the work with several of our colleagues in multiple fields and none of them complained about the technical hardship in understanding.

Reviewer #2

We thank the Reviewer 2 for the positive comments and constructive suggestions to our manuscript.

1. >> One concern has arisen from the new Fig 4 and corresponding SI Figure. An unspecified "control" is shown for each target site - this needs to be clearly defined. If "control" is untreated cells the authors should describe this. If so, it is surprising that some non-zero level of editing is observed in the control.

“Control” means untreated cells. The non-zero background mostly comes from the sequencing technique. It can be seen from other published papers as well, for example, from the Figure 3b in *Nature* 533(7603): 420–424 (2016). For clarity, we have changed the labels from “control” to “untreated” in Figure 4, and SI Figure 2 and 4.

2. >> Further, the data in SI Fig 4 is not clear that there is a significant improvement of editing selectivity (at least while maintaining on-target editing efficiency). This either needs to be addressed with a statistical test or the text should be updated to reflect this.

We performed the two-tailed unpaired t-test as suggested. The results indicate that the improvement of the editing selectivity by A3G3.15 over A3G3.1 is statistically significant for K562 cells ($p = 0.0005$) and Jurkat cells ($p = 0.0001$). The improvement for HeLa cells is less significant ($p = 0.3786$) and needs further optimization in the future. We have added this description to the main text in Page 9, as well as in the figure legend of Figure S4.